# The Spinning Voltage Influence on the Growth of ZnO-rGO Nanorods for Photocatalytic Degradation of Methyl Orange Dye

**Pierre G. Ramos** [1], **Clemente Luyo** [1], **Luis A. Sánchez** [1], **Enrique D. Gomez** [2] and **Juan M. Rodriguez** [1,*]

1   Center for the Development of Advanced Materials and Nanotechnology, Universidad Nacional de Ingeniería, Av. Túpac Amaru 210, Lima 15333, Peru; pierreramos1990@gmail.com (P.G.R.); cluyo@uni.edu.pe (C.L.); lasr_uni@hotmail.com (L.A.S.)
2   Department of Materials Science and Engineering, and Materials Research Institute, The Pennsylvania State University, University Park, PA 16802, USA; egomez9@gmail.com
*   Correspondence: jrodriguez@uni.edu.pe

**Abstract:** In this work, well-designed zinc oxide-reduced graphene oxide (ZnO-rGO) nanorods (NRs) were synthesized by a hydrothermal method using electrospun ZnO-rGO seed layers. The ZnO-rGO seed layers were fabricated on fluorine-doped tin oxide (FTO) glass substrates through calcined of electrospun nanofibers at 400 °C in the air for 1 h. The nanofibers were prepared by electrospinning different spinning voltages and a spinning solution containing zinc acetate, polyvinyl pyrrolidone, and 0.2 wt% rGO. From a detailed characterization using various analytical techniques, for instance, X-ray diffraction (XRD), field emission scanning electron microscopy (SEM), Raman spectroscopy, photoluminescence (PL), and X-ray photoelectron spectroscopy (XPS), the dependence of the structure, morphology, and optical properties of the ZnO-rGO NRs was demonstrated. The photocatalytic activities of ZnO-rGO nanorods were evaluated through the degradation of dye methyl orange (MO). The results show that the change of spinning voltages and the coupling of rGO with ZnO improved photodecomposition efficiency as compared to pure ZnO. The highest photocatalytic efficiency was obtained for the ZnO-rGO NRs prepared with a spinning voltage of 40 kV.

**Keywords:** reduced graphene oxide; ZnO nanorods; photocatalysis; spinning voltage

---

## 1. Introduction

Wastewater contaminants that include organic dyes, detergents, and pesticides, which originate in various industrial fields such as textile, plastic, or agriculture, have become one of the most serious environmental problems in recent years [1,2]. Various processes have been developed for the treatment of water contaminated with organic dyes, such as chemical oxidation, adsorption, ion-exchange, and photocatalytic treatments [3]. From these techniques, adsorption remains as the main technology used for these purposes [4–6]. However, this treatment method is less effective to cleanse organic pollutants at very low concentrations [7]. In contrast, photocatalysis which is a promising, simple, and eco-friendly technology, has been attractive to treat low-concentration organic contaminants in recent years [8]. So far, various semiconductor photocatalysts, such as ZnO, $TiO_2$, CuO, $Fe_2O_3$, and $WO_3$, have demonstrated promising photocatalytic activity [9–11], highlighting particularly zinc oxide due to its lower cost, non-toxicity, and size-tunable physicochemical properties [11,12].

Currently, the key challenge for improving the photocatalytic dye degradation efficiency of ZnO is to inhibit the recombination of photogenerated charge carriers (electron and holes). Thus, strategies like nanostructuring, doping, and formation of nanocomposites have been adopted to achieve

this improvement [13,14]; likewise, the photocatalytic activity of the photocatalyst can also be improved by enlarging its specific surface area through morphological modification [15]. Among the different types of nanostructures [16], 1D nanostructures like nanorods have received much attention due to their superior photocatalytic property [17,18]. In particular, these nanorods can be fabricated through hydrothermal-assisted electrospinning, as previously reported [19–21], which involves depositing a ZnO seed layer by electrospinning on a substrate and then growing ZnO nanorods. This method can be considered as an effective and easy method for fabrication of nanostructures.

Additionally, the formation of a nanocomposite with graphene-based materials improves the photocatalytic activity of ZnO due to more efficient charge separation. Specifically, reduced graphene oxide (rGO), offers new opportunities in photocatalysis as proved by several studies [22–24]. In addition, our previous work [25] demonstrated the influence of rGO amount on the fabrication and photocatalytic activity of zinc oxide-reduced graphene oxide (ZnO-rGO) nanorods, finding an optimal rGO content of 0.2 wt%. Thereby, in this work using this optimized amount of rGO, ZnO-rGO nanorods were synthesized on fluorine-doped tin oxide (FTO) glass substrates by low-temperature hydrothermal growth and electrospinning deposition applying different spinning voltages. Furthermore, we determined the effect of spinning voltages applied and rGO content on the morphology, crystallinity, optical and photocatalytic properties of the obtained ZnO-rGO nanorods. Detailed morphological, structural, and optical characterization of ZnO and ZnO-rGO nanorods were investigated by field emission scanning electron microscopy (FE-SEM), X-ray diffraction (XRD), Raman spectroscopy, and photoluminescence (PL). Surface properties and chemical composition of ZnO-rGO nanorods were analyzed by X-ray photoelectron spectroscopy (XPS). Finally, the photocatalytic methyl orange degradation performances of the ZnO-rGO nanorods (NRs) were studied.

## 2. Results and Discussion

### 2.1. SEM Analysis

Figure 1b–d shows the FE-SEM images of ZnO-rGO seed layers prepared by electrospinning using a spinning solution with 0.2 wt% rGO and applying different spinning voltages. The FE-SEM image of pristine ZnO seed layer is shown in Figure 1a. The regions enclosed by white lines indicated in Figure 1b–d confirm that ZnO nanoparticles were attached and grown on the surface of rGO sheets, due to covalent coupling/physisorption with the functional groups present in the reduced graphene oxide (-COOH,-OH and =O) [26,27], which play an important role in the formation of ZnO-rGO seeds by providing active sites for ZnO nanoparticles to nucleate and grow [28–30]. In addition, it can be seen from Figure 1 that the increase in the spinning voltage during electrospinning causes a decrease in the average size of nanoparticles that form the ZnO-rGO seeds compared to the size of particles that make up the pristine ZnO seed layers (20 ± 3 nm), in the following order: 13 ± 2 nm, 11 ± 1 nm, and 10 ± 1 nm for ZnO-rGO seed layers fabricated with a spinning voltage of 20 kV, 30 kV, and 40 kV, respectively. This result could be due to two reasons: (i) the presence of rGO in the spinning solution, which increased its electrical conductivity [31] and (ii) the increase of the spinning voltage [32]. Previous works have shown that these two reasons produce a decrease in the size of nanoparticles that make up the fabricated nanostructures by electrospinning [33,34].

The FE-SEM images for the pure ZnO and ZnO-rGO NRs obtained by a wet chemical method are shown in Figure 1e–h. The higher magnification FE-SEM images are shown as an inset in the lower left corner of Figure 1e–h and indicate that nanorods come together in a flower-like architecture. Furthermore, the coupling of rGO in the seed layers produces defective hexagonal nanorods grown in random orientations as demonstrated in previous work [25]. When only ZnO layers were deposited on FTO, the diameters of the pure ZnO nanorods obtained (Z20) were in the range 42–58 nm. The adhesion of the rGO and the spinning voltage increase produce a decrease in the dispersion of diameter size

and the mean diameter of the ZnO-rGO nanorods, with almost similar sizes. The average size of the ZnO-rGO NRs was: $34 \pm 6$ nm, $32 \pm 3$ nm, and $29 \pm 3$ nm for ZG20, ZG30, and ZG40, respectively.

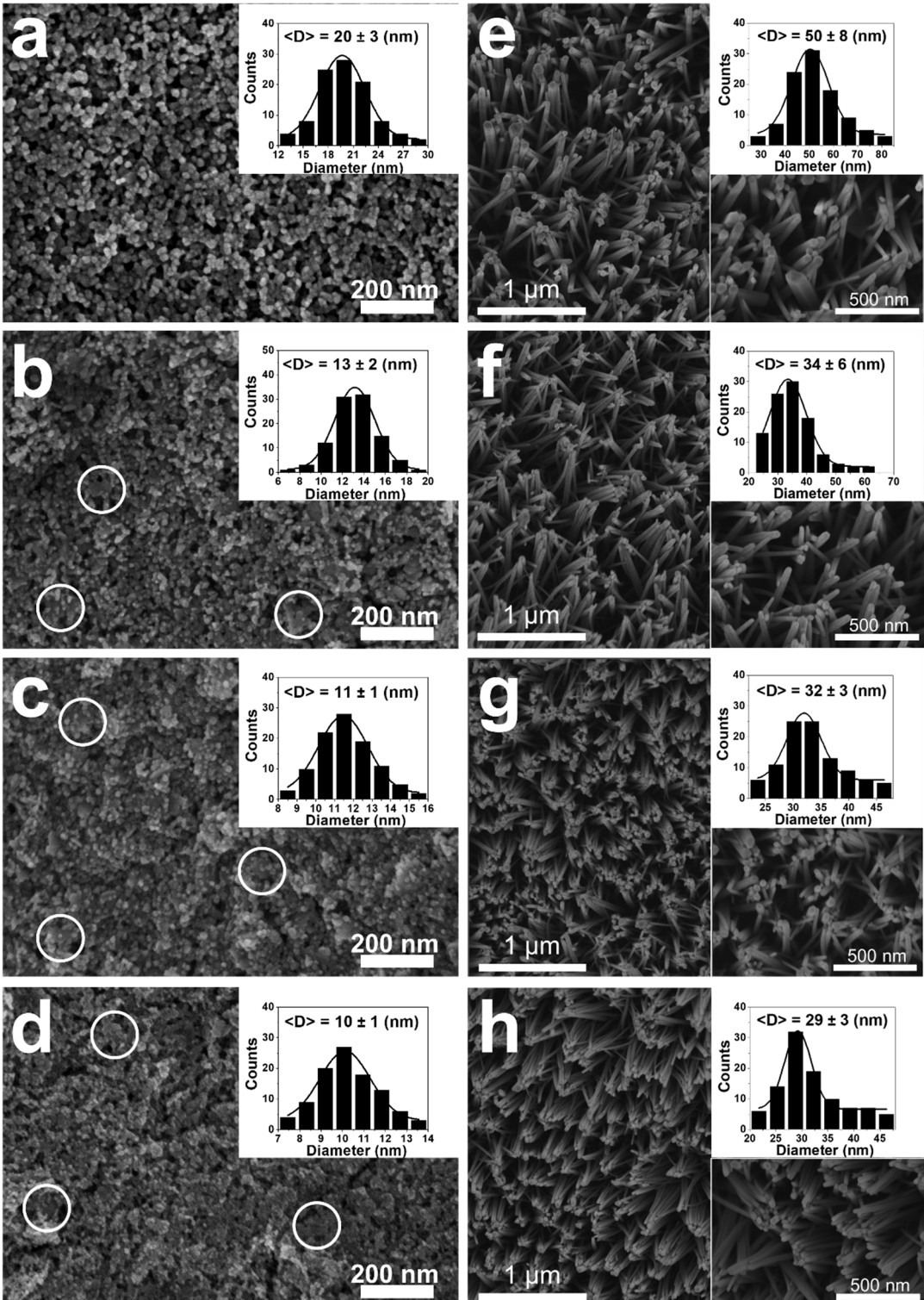

**Figure 1.** Field emission scanning electron microscopy (FE-SEM) images of seed layers (**a–d**) and top view of nanorods (**e–h**) labeled as Z20 (**a,e**), ZG20 (**b,f**), ZG30 (**c,g**), and ZG40 (**d,h**). The size distributions and corresponding standard deviations (SD) are shown as insets.

## 2.2. XRD Analysis

The crystal structure and phase purity of the ZnO-rGO NRs were characterized by X-ray diffraction analysis. The XRD patterns of obtained samples are shown in Figure 2. Six principal peaks from the diffractograms were obtained; these peaks can be ascribed to the (100), (002), (101), (102), (110), and (103) planes (JCPDS card No. 36-1451) of the hexagonal wurtzite structure of ZnO [35]. The (002) plane indicates that the c-axis of the wurtzite structure is the preferred growth direction for the ZnO-rGO NRs fabricated by electrospinning using different voltages. The high intensity of the diffraction patterns for ZnO-rGO NRs proves the high crystalline nature of the material. Furthermore, the diffraction pattern of reduced graphene oxide is shown as an inset in the top left corner of Figure 2. This image reveals two main peaks corresponding to rGO ascribed to their (100) and (002) planes [36], which are not distinguished in the diffraction patterns of ZnO-rGO NRs mainly due to the destruction of rGO sheets during the formation of the nanostructure and to the limited amount of rGO used [37,38]. Other diffraction peaks of impurity phases were not detected, indicating the phase pure formation of ZnO-rGO NRs.

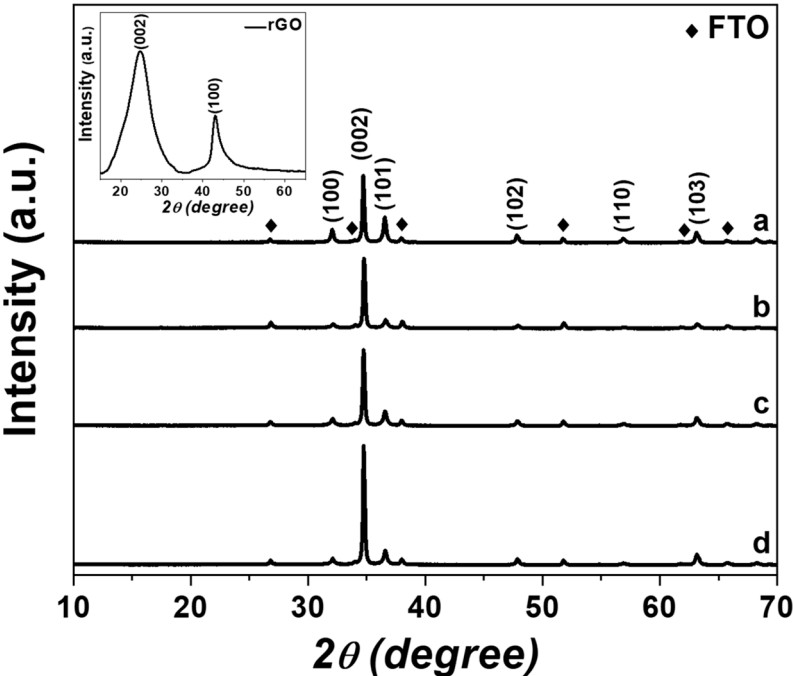

**Figure 2.** X-ray diffraction (XRD) patterns of (**a**) ZnO nanorods (NRs) and zinc oxide-reduced graphene oxide (ZnO-rGO) NRs grown from seed layers fabricated by electrospinning with 0.2 wt% rGO in the spinning solution using different spinning voltages: (**b**) 20 kV, (**c**) 30 kV, and (**d**) 40 kV. The XRD pattern of rGO is shown in the inset. The diffraction peaks corresponding to the fluorine-doped tin oxide (FTO) substrate are marked by "♦".

## 2.3. Raman Analysis

Analysis of Raman spectra was performed to verify the nature of our obtained nanorods. The Raman spectra of rGO, ZnO, and different ZnO-rGO samples are shown in Figure 3a. The intense peak at around 438 cm$^{-1}$ in ZnO Raman spectrum corresponds to E$_2$ (high) mode and is related to vibrations of the oxygen sublattice in ZnO [39], whereas second order peaks at 331 cm$^{-1}$ and 1142 cm$^{-1}$, also belonging to ZnO, were obtained [40,41]. The Raman spectrum of rGO features four peaks at 1348, 1587, 2714, and 2909 cm$^{-1}$ which are ascribed to the D, G, 2D, and D+G bands, respectively. The D band represents the sp$^3$ carbon defects [42], while the G band corresponds to the ordered sp$^2$ carbon network [43]. The 2D band can be related to the second order mode of the D band [44] and the D+G band represents an estimation of the disorder [42]. Raman spectra of ZnO-rGO NRs obtained by

electrospinning using different voltages show bands related to ZnO around at 438 cm$^{-1}$ (Figure 3b) and related to reduced graphene oxide at ~1366 cm$^{-1}$ and ~1593 cm$^{-1}$ for D and G band, respectively, which suggests the permanence of the carbonaceous material in the ZnO-rGO nanorods. Furthermore, a red-shift is observed in the G and D bands that confirm the formation of C-O-Zn linkage [45] in the ZnO-rGO nanorods due to the interaction between the oxide semiconductor and the carbonaceous material. The intensity ratio of D band ($I_D$) to G band ($I_G$) correlates with the disorder evaluation in carbonaceous materials [46]. The values of $I_D/I_G$ are 0.84, 1.01, 1.06, and 1.00 for rGO, ZG20, ZG30, and ZG40 samples, respectively. The larger $I_D/I_G$ ratio of ZnO-rGO NRs compared to rGO suggests the creation of more defects in the ZnO-rGO NRs obtained by hydrothermal treatment caused by the presence of zinc oxide [46,47].

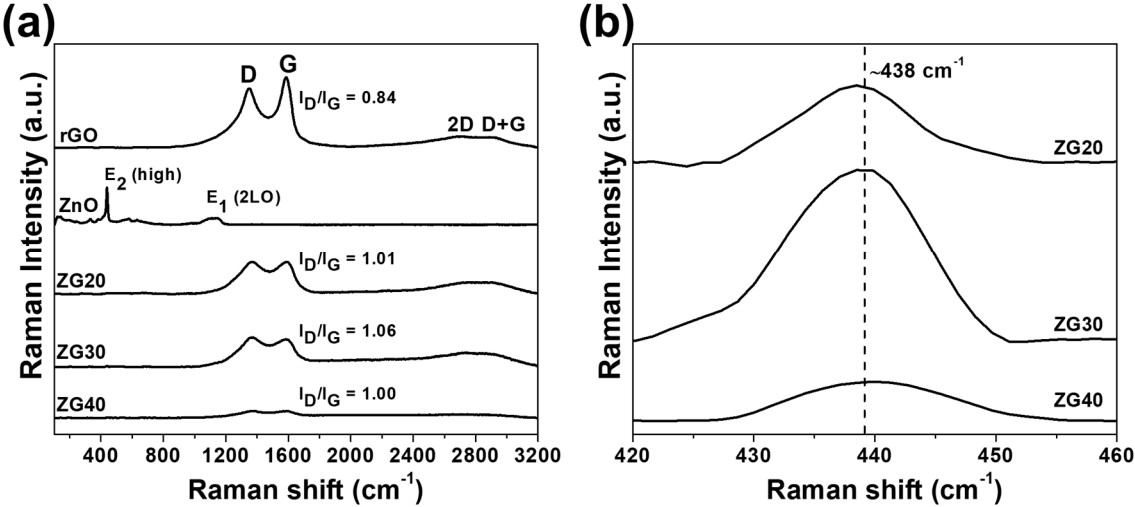

**Figure 3.** (**a**) Raman spectra of rGO, ZnO NRs, and the ZnO-rGO (ZG20, ZG30, ZG40) NRs; and (**b**) enlarged Raman spectrums at 420–460 cm$^{-1}$ of ZnO-rGO (ZG20, ZG30, ZG40) NRs grown from seed layers fabricated by electrospinning with 0.2 wt% rGO in the spinning solution using different spinning voltages.

*2.4. PL Analysis*

Measuring photoluminescence (PL) is an effective means of investigating the separation process of e$^-$ and h$^+$ pairs in semiconductors. Figure 4a shows the PL spectra with a 325 nm excitation wavelength of pure ZnO and ZnO-rGO NRs, fabricated by electrospinning using different spinning voltages and a spinning solution with 0.2 wt% rGO. It was found that all samples have the same emission shape with three emission peaks located around 385, 600, and 760 nm. Enlarged PL spectrums of ZnO and ZnO-rGO NRs in the range at 370–400 nm wavelengths are shown in Figure 4b. Near-band edge (NBE) emission around 385 nm (shown in Figure 4b) is attributed to the free exciton recombination process [48]. Furthermore, a blue-shift in this peak was observed for ZnO-rGO NRs compared to ZnO NRs. These shifts as demonstrated by Jayalakshmi et al. [49] are due to the presence of reduced graphene oxide in the fabricated ZnO-rGO NRs. The emission peaks in the visible region (~600 nm) and the near-infrared region (~760 nm) were assigned respectively to structural/intrinsic defects and impurities in the structures [50,51], and the second-order diffraction of the NBE emission band [52]. Moreover, the spinning voltages used for the fabrication of ZnO-rGO NRs and the presence of rGO in the samples produced a significant decrease in intensities of ZnO emission peaks as follows, $I_{ZnO} > I_{ZG20} > I_{ZG30} > I_{ZG40}$, which is associated with the decreasing in the diameters of NRs (see Figure 1) due to the increase of spinning voltages and the presence of rGO in the nanostructures [53,54]. This decrease in PL intensity indicates a reduction of electron-hole recombination in the ZnO-rGO samples and enhancement of charge transfer at the interface [52,54], which can improve the photocatalytic activity of the obtained nanorods.

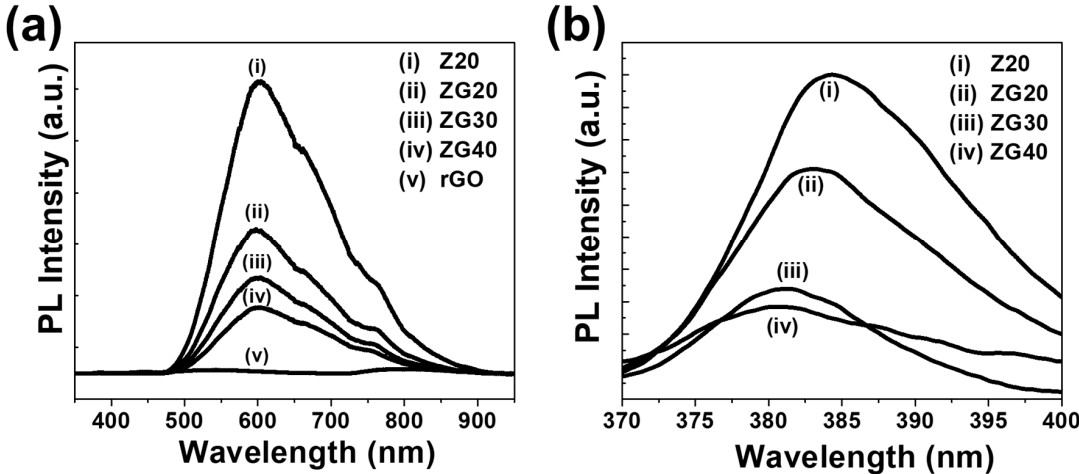

**Figure 4.** (**a**) Photoluminescence (PL) spectra and (**b**) enlarged PL spectrums at 370–400 nm wavelengths of ZnO (Z20) and ZnO-rGO (ZG20, ZG30, ZG40) NRs grown from seed layers fabricated by electrospinning with 0.2 wt% rGO in the spinning solution using different spinning voltages.

*2.5. XPS Analysis*

The chemical composition on the surface and the state of elements present on ZnO and ZnO-rGO NRs have been characterized by X-ray photoelectron spectroscopy (XPS). Figure 5a shows the XPS survey spectra for the obtained nanorods and reveals only the presence of zinc (Zn), oxygen (O), and carbon (C) elements. In addition, it can be seen that the spectra of the ZnO and ZnO-rGO NRs contain C1s (284.53 eV), O1s (529.33 eV), and Zn2p (1020.53 eV) peaks. The C1s peak also found in Z20 full spectrum is because the pure ZnO NRs are exposed to the atmosphere where carbon contamination can take place [55]. Figure 5b presents the high-resolution $Zn2p_{3/2}$ XPS spectra measured for all ZnO-rGO NRs. The obtained results are consistent with the oxidation state of $Zn^{2+}$ in ZnO [56,57]. A remarkable shift of ZnO-rGO NRs can be seen in $Zn2p_{3/2}$ region (Figure 5b) in comparison with the pure ZnO NRs, caused by a strong interaction between zinc oxide and reduced graphene oxide [58,59]. This interaction produces a good separation of the photo-induced charge carriers. Thus, the performance of the photocatalyst was enhanced [58,59].

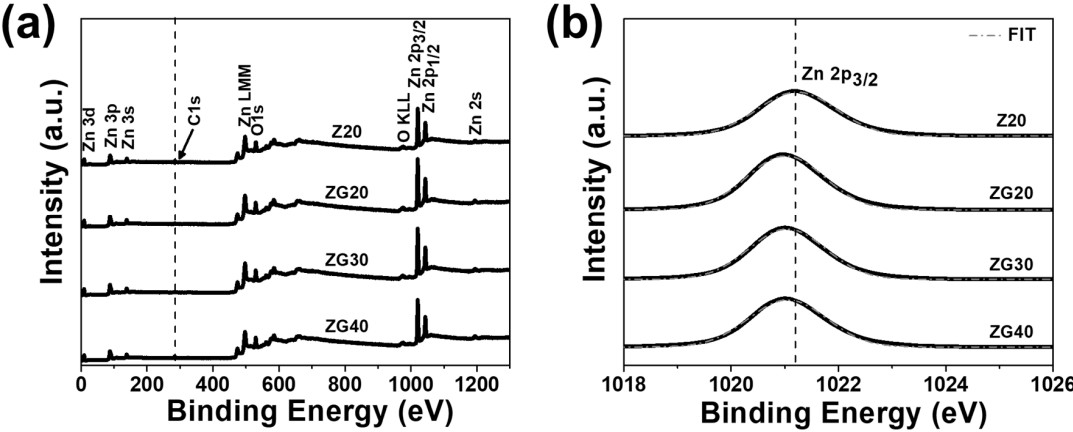

**Figure 5.** (**a**) X-ray photoelectron spectroscopy (XPS) survey spectra and (**b**) $Zn2p_{3/2}$ spectra of Z20, ZG20, ZG30, and ZG40 nanorods.

The high-resolution scan of C1s (left side) and O1s (right side) for ZnO-rGO NRs is shown in Figure 6. First, the C1s XPS spectrum, acquired from 281 to 292 eV, was deconvoluted into five characteristic peaks, labeled as $C_1$, $C_2$, $C_3$, $C_4$, and $C_5$. The peak at around 282.4 eV ($C_1$) corresponds to the C-O-Zn bond [58]. The binding energy at around 284.8 eV ($C_2$) corresponds to the C-C of rGO,

while the other three peaks located approximately at 286.4, 288.5, and 292 eV are assigned to C-O/C-OH ($C_3$), C=O ($C_4$), and O-C=OH ($C_5$) functional groups, respectively [60,61]. The XPS high-resolution O1s spectra are shown in Figure 6 (right side), which were acquired from 525 to 539 eV and deconvoluted into four oxygen species, denoted as $O_1$, $O_2$, $O_3$, and $O_4$. The binding energy peaks obtained at 529.7, 531.0, 533.6, and 536.7 eV, corresponding to $O^{2-}$ ions of the ZnO lattice ($O_1$) [62], oxygen deficiencies lattice in ZnO ($O_2$) [48], oxygen species chemisorbed on the surface ($O_3$) [63], and oxygen in the O-C=O bond ($O_4$) [57,64], respectively.

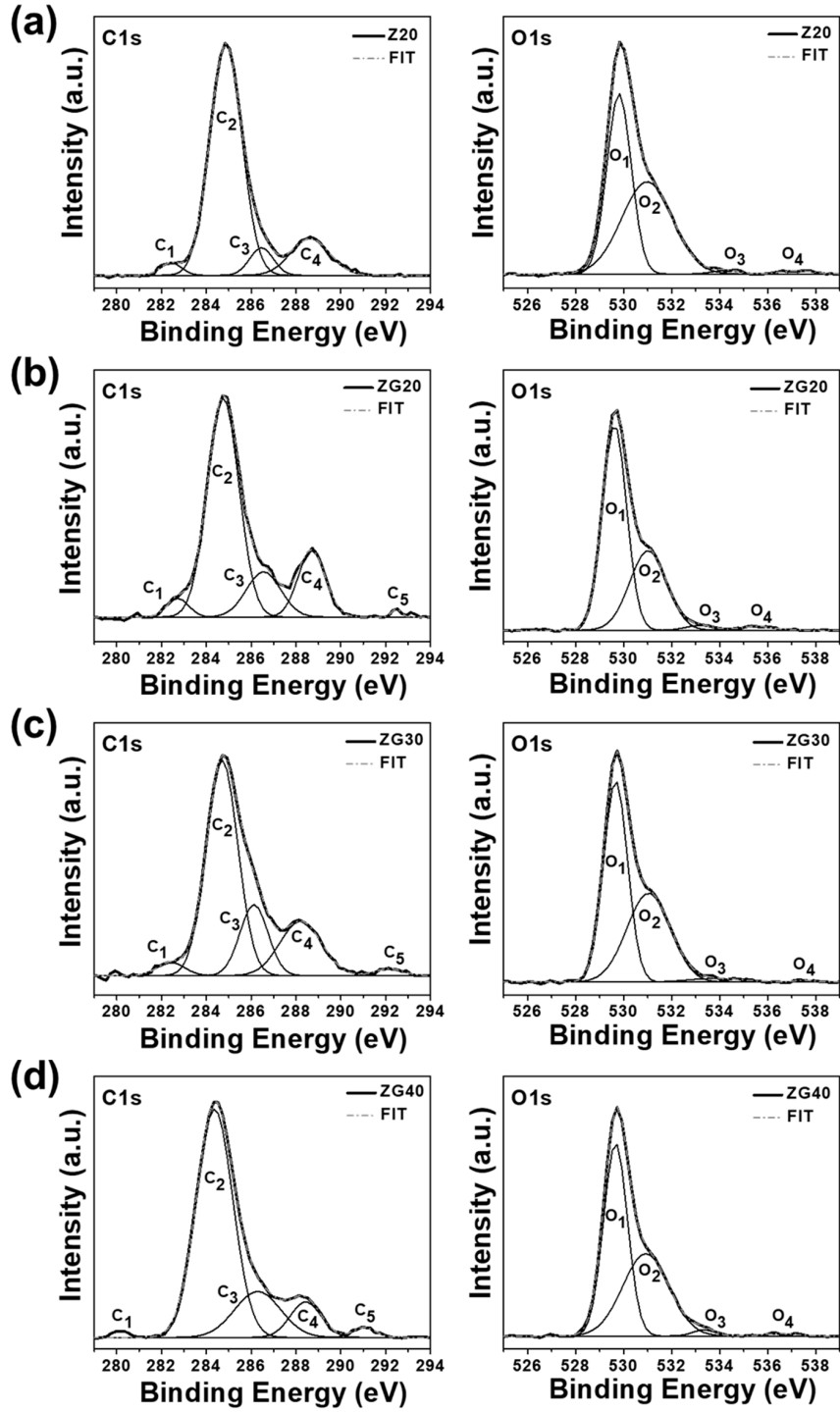

**Figure 6.** High-resolution C1s (**left side**) and O1s (**right side**) spectrum of (**a**) Z20, (**b**) ZG20, (**c**) ZG30, and (**d**) ZG40 nanorods.

## 2.6. Photocatalytic Decolorization of Methyl Orange

The extent of adsorption depends on the physicochemical characteristics of the catalyst and pollutant [65]. Hence, methyl orange (MO) dye adsorption experiment was carried out in the presence of catalyst and the absence of light. This experiment showed no significant adsorption of MO onto the catalyst. In addition, as a reference we applied light irradiation on methyl orange in the absence of a photocatalyst; no degradation of dye was observed, corroborating that the degradation was effectively driven by a photocatalytic process. The photocatalytic dye degradation performances of the ZnO and ZnO-rGO NRs were evaluated in aqueous solution of methyl orange dye (5 ppm) under UV light irradiation, and the results are plotted in Figure 7a. Figure 7a shows the change in the methyl orange concentration in aqueous solution in the presence of ZnO-rGO NRs as a function of irradiation time. $C$ is the concentration of MO at the irradiation time ($t$) and $C_0$ represents the initial concentration of MO. The results indicate that ZnO-rGO NRs synthesized by electrospinning, applying different spinning voltages and a spinning solution with 0.2 wt% rGO can improve the photocatalytic activity of single ZnO NRs. The ZG40 photocatalyst shows the highest photocatalytic activity compared with Z20, ZG20, and ZG30 photocatalysts. The degradation efficiency of the photocatalyst based on ZG40 NRs shows a maximum degradation of ~99% at 6 h, whereas the degradation efficiency of Z20, ZG20, and ZG30 is ~77%, ~95%, and ~97% at 7 h of irradiation time, respectively. The enhancement of photocatalytic performance of ZnO-rGO NRs as compared to pristine ZnO NRs is mainly attributed to the fact that the photogenerated electrons from ZnO excited by a light source are trapped by the reduced graphene oxide, avoiding recombination of electron-hole pairs [66], and also to the increase in specific surface area that in turn relies on the decrease in particle size (see Figure 1), caused by the increase of voltage during electrospinning [67,68]. According to the XPS analysis, it can be seen that the peak's intensity at 288.5 eV ($C_3$) of ZnO-rGO NR photocatalysts is much stronger than that of pure ZnO, indicating the significant increase of hydroxy groups on the surface of ZnO-rGO NRs, compared with pure ZnO NRs. The hydroxy groups can act as adsorption centers on which the degradation of pollutants takes place and are helpful to enhance the photocatalytic activity of ZnO-rGO NRs [69,70]. Moreover, a high amount of oxygen vacancies was found on the surface of the ZG40 sample. The presence of oxygen-deficient centers on the surface can reduce the rate of electron-hole pair recombination [69–71]. Thereby, the highest photocatalytic activity was obtained for the ZnO-rGO NRs fabricated using 40 kV of spinning voltage as was demonstrated by the photocatalytic experiment.

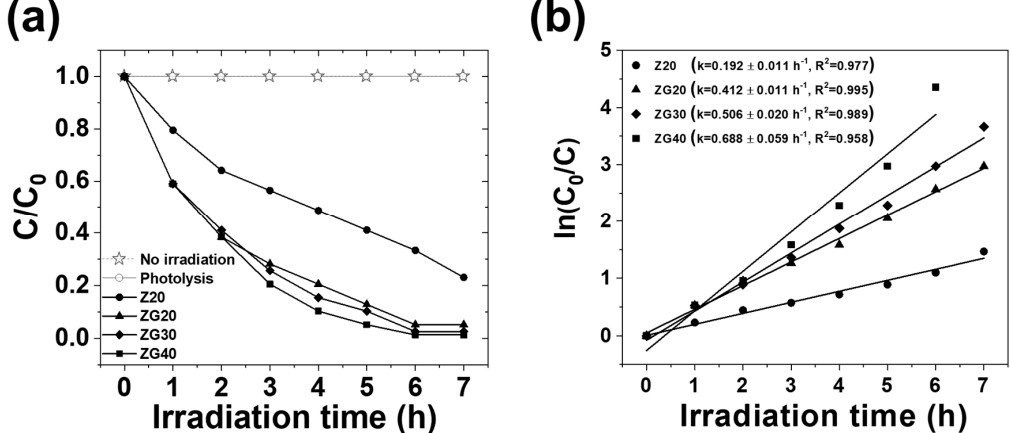

**Figure 7.** (**a**) Photodegradation curves of methyl orange with different ZnO-rGO NR photocatalysts. (**b**) Kinetics plots calculated from (**a**) of ZnO-rGO NR photocatalysts.

The MO degradation process was further investigated by the first-order reaction kinetic model [72]. The rate expression is: $ln(C_0/C_t) = kt$, where $C_0$ is the initial absorbance, $C_t$ is the absorbance after time $t$, and $k$ is the first-order rate constant. A plot between $ln(C_0/C)$ and reaction time is shown in Figure 7b. The estimated degradation rate constants for the Z20, ZG20, ZG30, and ZG40 nanorods were

found to be $0.192 \pm 0.011 \ h^{-1}$, $0.412 \pm 0.011 \ h^{-1}$, $0.506 \pm 0.020 \ h^{-1}$, and $0.688 \pm 0.059 \ h^{-1}$, respectively. Furthermore, a linear dependence between this kinetic parameter and the spinning voltage was found. The high values obtained for ZnO-rGO nanorods compared to ZnO signify an improvement of the photocatalytic activity due to the increase of voltage during spinning fabrication and by the incorporation of rGO into ZnO seed layers. The calculated values of the correlation coefficient ($R^2$) for Z20, ZG20, ZG30, and ZG40 nanorods were 0.977, 0.995, 0.989, and 0.958, respectively, which are close to 1. It demonstrates that the degradation process of MO through ZnO and ZnO-rGO NRs follows first-order reaction kinetic. The results obtained for the photocatalytic efficiency of the nanorods obtained are in good agreement with the results obtained by photoluminescence (see Figure 4).

The improvement in the photocatalytic degradation efficiency of the nanorods fabricated in this work is related to the increase in spinning voltages and the rGO adherence into the ZnO-rGO seed layers. The possible photocatalytic degradation mechanism of MO is proposed for ZnO-rGO NRs, as illustrated in Figure 8. When the ZnO nanostructure is irradiated from the light source, the excited electrons ($e^-$) migrate from the valence band (VB) of ZnO to its conduction band (CB) with generation simultaneous to the same number of holes ($h^+$) in the VB (Equation (2)). The rGO sheets receive these photo-excited electrons from the conduction band of ZnO nanorods and act as an electron transporter phase hindering the electron-hole recombination [55]. This is mainly because the energy levels of rGO and ZnO are different from each other. The value of the work function of rGO is $-4.42$ eV, which is lower than the value of $-4.05$ eV obtained for the ZnO conduction band [67]. The separated $e^-$–$h^+$ pairs react with oxygen molecules and with the surface absorbed water molecules as described in Equations (1)–(7), producing the strongly oxidizing hydroxyl radical (OH) and oxygen radical anion $\left(O_2^-\right)$, which react with the MO dye molecules in solution and degrade them [73,74]. Based on these results we conclude that the variation of spinning voltages and the presence of rGO enhance the photocatalytic activity performance of pure ZnO nanorods, suppressing photoinduced charge recombination effectively. Finally, the overall chemical and photocatalytic reactions are shown below:

$$ZnO/rGO + h\nu \rightarrow ZnO/rGO\left(e_{CB}^- + h_{VB}^+\right) \tag{1}$$

$$ZnO + h\nu \rightarrow ZnO\left(e_{CB}^- + h_{VB}^+\right) \tag{2}$$

$$ZnO(e_{CB}^-) \rightarrow rGO(e_{tr}^-) \tag{3}$$

$$ZnO/rGO(h_{VB}^+) + OH^- \rightarrow ZnO/rGO + OH \tag{4}$$

$$rGO\left(e_{tr}^-\right) + O_{2(ads)} \rightarrow O_2^- \tag{5}$$

$$O_2^- + pollutent\ (MO) \rightarrow decomposed\ products \tag{6}$$

$$OH + pollutent\ (MO) \rightarrow decomposed\ products \tag{7}$$

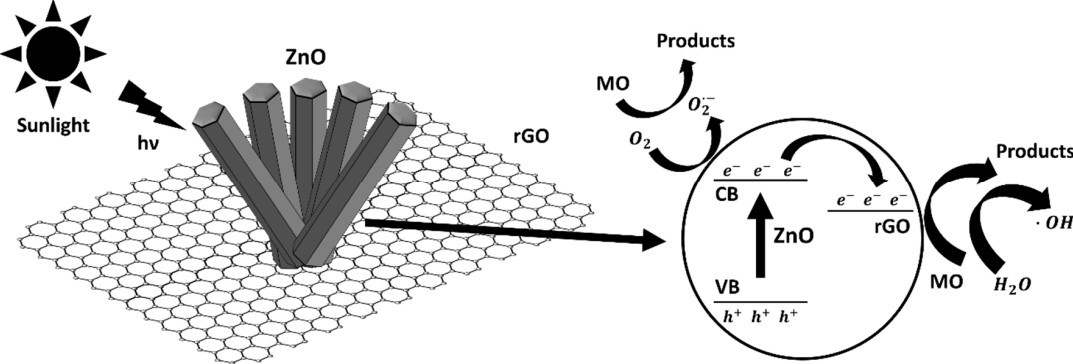

**Figure 8.** A schematic diagram for the charge-transfer process and photocatalytic activity of ZnO-rGO NRs for methyl orange (MO) degradation.

## 3. Materials and Methods

### 3.1. Materials, Reagents, and Chemicals

In this work, zinc acetate dihydrate ($Zn(CH_3COO)_2$ $2H_2O$, Merck, Darmstadt, Germany), polyvinylpyrrolidone (PVP, Sigma-Aldrich, St. Louis, MO, USA), and N,N-dimethylformamide ($HCON(CH_3)_2$, Merck, Darmstadt, Germany) were used as precursors of ZnO seed layers, while commercial reduced graphene oxide (Sigma-Aldrich, St. Louis, USA) was additionally used to obtain the ZnO-rGO seed layers. In addition, zinc nitrate hexahydrate ($Zn(NO_3)_2$ $6H_2O$, Sigma-Aldrich, Steinheim, Germany) and sodium hydroxide (NaOH, Merck, Darmstadt, Germany) were employed in the preparation of ZnO and ZnO-rGO nanorods growth solutions. All of the chemicals used to obtain the samples were used as received.

FTO glass conductive plates of 3.5 cm × 2 cm of area and 7 $\Omega/cm^2$ of resistance were used as substrates for the deposition of ZnO and ZnO-rGO nanorods.

### 3.2. Preparation of ZnO and ZnO-rGO Nanorods

ZnO and ZnO-rGO seed layers were deposited onto FTO glass plates using the electrospinning technique, and subsequently, they were used as substrates to grow ZnO and ZnO-rGO NRs, respectively, by hydrothermal treatment. To obtain a pure ZnO seed layer, a spinning solution containing 1 g of zinc acetate dihydrate and 1 g of polyvinylpyrrolidone dissolved in N-N dimethylformamide was used. ZnO-rGO seed layers were obtained by using the same spinning solution seen above, adding now 0.2 wt% rGO and varying the spinning voltage in three values: 20 kV, 30 kV, and 40 kV. The obtained spinning solutions were loaded into a plastic syringe fitted with a 0.6 mm diameter needle made of stainless steel. The syringe supplied the feeding solution at a speed of 2 mL $h^{-1}$ with a deposition time of 3 h. Then, the ZnO-rGO seed layers were obtained by calcination in a muffle furnace at 400 °C of the coated substrates. The solution medium used for the growth of the ZnO and ZnO-rGO NRs was prepared according to Rodriguez et al. [75] by mixing equal volumes of zinc nitrate hexahydrate (0.15 M) and sodium hydroxide (2.1 M) in water. Then, the substrates seeded with ZnO and ZnO-rGO films were placed in 100 mL screw-capped glass flasks containing 40 mL of the solution for the growth. These glass flasks containing the substrates and the solutions for the growth were placed in an oven at 90 °C for 1 h. After that, the substrates covered with ZnO and ZnO-rGO NRs were removed from the solution, cleaned with distilled water and ethanol and finally dried at 70 °C. ZnO nanorods grown from ZnO seeds prepared with 20 kV of spinning voltage were labeled as Z20, whereas the ZnO-rGO nanorods grown from ZnO-rGO seeds prepared with 20 kV, 30 kV, and 40 kV of spinning voltages were labeled as ZG20, ZG30, and ZG40, respectively. Figure 9 illustrates the formation procedure of ZnO-rGO nanorods arrays.

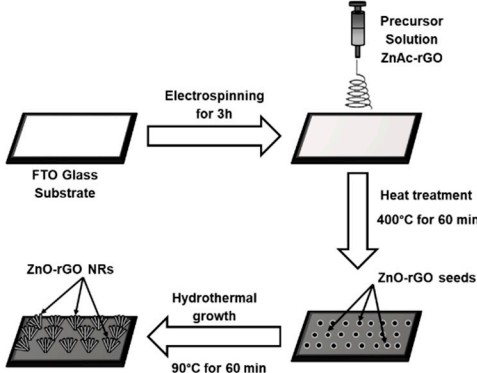

**Figure 9.** Schematic diagram showing the process applied to produce the ZnO-rGO nanorod arrays.

*3.3. Photocatalytic Characterization*

The photocatalytic activity of ZnO and ZnO-rGO nanorods was measured through degradation of an aqueous solution of methyl orange (MO) using a 220 W OSRAM Ultravitalux lamp (OSRAM, Wilmington, DE, USA) placed approximately at 20 cm from the system, where 70 Wm$^{-2}$ in the UV-A range of intensity was measured. The initial concentration of MO was 5 ppm. In the typical experiment, 3 mg of ZnO and ZnO-rGO nanorods were added in 50 mL MO in a 100 mL beaker. Experiments were performed at room temperature of 23 °C ± 2 °C. During the irradiation, 3 mL of the treated solution was collected each hour in order to analyze the methyl orange concentration by a Lambda 25 UV–Vis spectrophotometer (PerkinElmer, Waltham, MA, USA) at 462 nm. The photocatalytic activity of the ZnO and ZnO-rGO NRs fabricated with different spinning voltages during the electrospinning process was compared.

*3.4. Characterization Methods*

The obtained nanostructures were characterized by X-ray diffraction using a Bruker D8 Advance (Bruker, Billerica, MA, USA). The 2$\theta$ range was from 10 to 70 degrees with CuK$\alpha$ radiation ($\lambda$ = 1.5418 Å). The morphologies of ZnO-rGO seeds as well as of ZnO-rGO NRs were visualized by field emission scanning electron microscope HITACHI SU8230 (Hitachi, Omuta, Japan). ImageJ 1.52r software was used in order to obtain the size distributions of nanoparticles and fabricated nanorods. Room temperature Raman and photoluminescence spectroscopy were obtained on a Renishaw inVia equipment (Renishaw, Wotton-under-Edge, UK) equipped with a 514 nm wavelength of Ar laser light and by He-Cd laser source with a wavelength of 325 nm, respectively. The Raman spectrums were recorded over a range of 100 cm$^{-1}$ to 3200 cm$^{-1}$, while the photoluminescence measurements were recorded over the wavelength range of 350 nm to 950 nm. X-ray photoelectron spectroscopy was utilized to identify chemical states of prepared samples by using a Physical Electronics VersaProbe II (Physical Electronics, Chanhassen, MN, USA) instrument equipped with a monochromatic Al (K$\alpha$) X-ray source (h$\nu$ = 1486.7 eV) and a concentric hemispherical analyzer.

## 4. Conclusions

In summary, a series of ZnO and ZnO-rGO NRs were synthesized on ZnO and ZnO-rGO seed layers by a wet chemical method, and the seed layers were obtained by electrospinning using a precursor solution with 0.2 wt% rGO and applying different spinning voltages. Furthermore, the photocatalytic methyl orange degradation performances were evaluated. The obtained results show an effective formation of ZnO-rGO NRs, where the adhered rGO sheets and different spinning voltages applied influenced the structure, morphology, photocatalytic performance, and optical properties of the nanorod samples. Significant enhancement of the photocatalytic efficiency of fabricated ZnO-rGO nanorods as compared with pure ZnO nanorods was determined. The ZnO-rGO nanostructure fabricated with 40 kV of spinning voltage exhibited the highest photocatalytic activity for dye photodegradation. The enhancement in photocatalytic activity of ZnO-rGO NRs is attributed to the higher transfer rate of photo-generated electrons from ZnO to rGO, the high efficiency in light utilization, and inhibited recombination of the photoinduced hole-electron pairs of ZnO. Moreover, electrospinning is a convenient method, compared with traditional methods, for the large-scale fabrication of ZnO-rGO nanorods arrays; we speculate that this approach can be extended further for the growth of other nanomaterials on substrates.

**Author Contributions:** Conceptualization, J.M.R. and P.G.R.; methodology, P.G.R.; investigation, P.G.R. and L.A.S.; resources, L.A.S., C.L., and E.D.G.; writing—original draft preparation, P.G.R.; writing—review and editing, E.D.G. and J.M.R.; visualization, C.L.; supervision, J.M.R.; funding acquisition, J.M.R. All authors have read and agreed to the published version of the manuscript.

**Funding:** This research was funded by the Peruvian National Fund for Scientific, Technological Development and Technological Innovation (FONDECYT), grant numbers 168-2015-FONDECYT and 32-2019-FONDECYT-BM-INC.INV, P.G.R, E.D.G. and J.M.R. want to thanks to the PSU-UNI-PUCP seed projects program for support.

**Acknowledgments:** The work described in this paper was financially supported by the projects INNOVATE (project number 113-INNOVATE PERU-ISASS-2018) and FINCYT (project number 133-FINCYT-IB-2015). P.G.R. wants to thanks the Ministry of Education of Peru for the PhD scholarship. The authors thanks Jeff Shallenberger of the Materials Characterization Lab at Penn State for his excellent technical assistance with X-ray photoelectron spectroscopy.

**Conflicts of Interest:** The authors declare no conflict of interest.

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
