# Peer review of "The Spinning Voltage Influence on the Growth of ZnO-rGO Nanorods for Photocatalytic Degradation of Methyl Orange Dye"

_catalysts, doi:10.3390/catal10060660_

Round 1
Reviewer 1 Report
In this work, Ramos and co-workers studied the effect of the voltage on the properties of ZnO-rGO nanorods prepared by electrospinning on the photocatalytic degradation of methyl orange dye. The work needs major revision before publication, as it follows reported.
1) Adsorption is still the main technology used for dye removal from wastewater. For example, the following paper may be briefly discussed or cited in the introduction: Li et al., Bioresource Technol. 2019, 277, 157-170; Vuono et al., Chinese J. Chem. Eng. 2017, 25, 523-532; Chaukura et al., Appl. Water Sci. 2017, 7, 2175-2186.
2) The authors report that 1.0 mL of rGO was used. But, the amount shoud be expressed in specific units, e.g. mL/g, wt%, and so on.
3) Figure 1: The caption should be improved. The authors may add labels on the figures.
4) The effect of surface properties on adsorption phenomena depends on the chemical-physical properties of the pollutants. In the discussion part, the authors should highlight this aspect. Why the authors choose methyl orange dye?
5) The authors should add details about the quality of the first-order reaction kinetic model, e.g. R2, standard deviation of the estimated parameters. Furthermore, the estimated kinetic parameters seem linearly dependent on voltage.
Author Response
REVIEWER #1
1) Adsorption is still the main technology used for dye removal from wastewater. For example, the following paper may be briefly discussed or cited in the introduction: Li et al., Bioresource Technol. 2019, 277, 157-170; Vuono et al., Chinese J. Chem. Eng. 2017, 25, 523-532; Chaukura et al., Appl. Water Sci. 2017, 7, 2175-2186.
AUTHOR RESPONSE:
On page 1, the introduction was modified from lines 34 to 39, incorporating the references suggested. The following sentences were added to introduction:
“From these techniques, the adsorption remains as the main technology used for these purposes [4-6]. However, this treatment method is less effective to cleanse the organic pollutants at very low concentrations [7]. In contrast, photocatalysis, which is a promising, simple and eco-friendly technology, has been attractive to treat low-concentration organic contaminants in recent years [8].”
2) The authors report that 1.0 mL of rGO was used. But, the amount should be expressed in specific units, e.g. mL/g, wt%, and so on.
AUTHOR RESPONSE:
1.0 mL of rGO was replaced by 0.2 %wt rGO in whole manuscript. The changes are shown in lines 18, 58-59, 72, 124, 130, 157-158, 163, 231, 308 and 349.
3) Figure 1: The caption should be improved. The authors may add labels on the figures.
AUTHOR RESPONSE:
Figure 1 was modified and showed in line 100. This change in the Figure produced that section 2.1 (lines 70-99) and the figure 1 legend (lines 102-107) were modified as well.
4) The effect of surface properties on adsorption phenomena depends on the chemical-physical properties of the pollutants. In the discussion part, the authors should highlight this aspect. Why the authors choose methyl orange dye?
AUTHOR RESPONSE:
In section 2.6 the following paragraph (lines 217-222) were added:
“The extent of adsorption depends on the physicochemical characteristics of the catalyst and pollutant [66]. Hence, methyl orange (MO) dye adsorption experiment was carried out in the presence of catalyst and the absence of light. This experiment showed no significant adsorption of MO onto the catalyst. Besides, as a reference we applied light irradiation on methyl orange in absence of photocatalyst; no degradation of dye was observed, corroborating that the degradation was effectively driven by a photocatalytic process.”
In addition:
Methyl orange was chosen in this work because of is an azo dye characterized by nitrogen to nitrogen double bonds (-N=N-) in their chemical structure, which are the most active bonds in azo-dye molecules and their cleavage leads to decoloration of dye. Besides, azo-dyes account for up to 70% of the dyes in use nowadays, hence the effort to degrade them.
5) The authors should add details about the quality of the first-order reaction kinetic model, e.g. R2, standard deviation of the estimated parameters. Furthermore, the estimated kinetic parameters seem linearly dependent on voltage
AUTHOR RESPONSE:
In section 2.6 (lines 254-256) the standard deviation of degradation rate constants was added. Moreover, the values of correlation coefficient (R2) were added in lines 258-260.
Finally, the kinetic parameters were added in Figure 7b. Thus, Figure 7 was modified and showed in line 264.

Reviewer 2 Report
Please see the attached file.

Author Response
REVIEWER #2
1) Nanorods have been formed using wet chemical process. The details of the process are not provided in the paper that are essential for reproducibility by other researchers.
AUTHOR RESPONSE:
The detailed process for growing nanorods was added to section 3.2 (lines 312-317):
The solution medium used for the growth of the ZnO and ZnO-rGO NRs was prepared according to Rodriguez et al. [76] by mixing equal volumes of zinc nitrate hexahydrate (0.15 M) and sodium hydroxide (2.1 M) in water. Then, the substrates seeded with ZnO and ZnO-rGO films were placed in 100 mL screw-capped glass flasks containing 40 mL of the solution for the growth. These glass flasks containing the substrates and the solutions for the growth were placed in an oven at 90 â—¦C for 1 h. After that, the substrates covered with ZnO and ZnO-rGO NRs were removed from the solution, cleaned with distilled water, ethanol, and finally dried at 70°C.
In addition, this section was modified in lines 304-306.
2) Figure 1 shows FE-SEM images of the ZnO-rGO. On the right side, the error in Fig. 1 (b – d) is big that it covers all the reported values from 34 to 29 nm. Please justify the error. Otherwise, the size of the rods is almost similar to each other in the images.
AUTHOR RESPONSE:
Is right. For that reason, we modified section 2.1 for explaining the dispersion of diameters and the average sizes of the nanorods obtained, adding the following sentences in lines 93-98: “When only ZnO layers were deposited on FTO, the diameters of the pure ZnO nanorods obtained (Z20) were in the range 42-58 nm. The adhesion of the rGO and the spinning voltage increase produce a decrease in the dispersion of diameters size and the mean diameter of the ZnO-rGO nanorods, with almost similar sizes”.

Round 2
Reviewer 1 Report
The manuscript was improved, it is now suitable for publication.